# A prospective study of consecutive emergency medical admissions to compare a novel automated computer-aided mortality risk score and clinical judgement of patient mortality risk

Muhammad Faisal,[1] Binish Khatoon,[1] Andy Scally,[2] Donald Richardson,[3] Sally Irwin,[4] Rachel Davidson,[4] David Heseltine,[4] Alison Corlett,[4] Javed Ali,[4] Rebecca Hampson,[4] Sandeep Kesavan,[4] Gerry McGonigal,[4] Karen Goodman,[4] Michael Harkness,[4] Mohammed Mohammed[1,5]

For numbered affiliations see end of article.

**Correspondence to**
Dr Mohammed Mohammed;
m.a.mohammed5@bradford.ac.uk

## ABSTRACT

**Objectives** To compare the performance of a validated automatic computer-aided risk of mortality (CARM) score versus medical judgement in predicting the risk of in-hospital mortality for patients following emergency medical admission.

**Design** A prospective study.

**Setting** Consecutive emergency medical admissions in York hospital.

**Participants** Elderly medical admissions in one ward were assigned a risk of death at the first post-take ward round by consultant staff over a 2-week period. The consultant medical staff used the same variables to assign a risk of death to the patient as the CARM (age, sex, National Early Warning Score and blood test results) but also had access to the clinical history, examination findings and any immediately available investigations such as ECGs. The performance of the CARM versus consultant medical judgement was compared using the c-statistic and the positive predictive value (PPV).

**Results** The in-hospital mortality was 31.8% (130/409). For patients with complete blood test results, the c-statistic for CARM was 0.75 (95% CI: 0.69 to 0.81) versus 0.72 (95% CI: 0.66 to 0.78) for medical judgements (p=0.28). For patients with at least one missing blood test result, the c-statistics were similar (medical judgements 0.70 (95% CI: 0.60 to 0.81) vs CARM 0.70 (95% CI: 0.59 to 0.80)). At a 10% mortality risk, the PPV for CARM was higher than medical judgements in patients with complete blood test results, 62.0% (95% CI: 51.2 to 71.9) versus 49.2% (95% CI: 39.8 to 58.5) but not when blood test results were missing, 50.0% (95% CI: 24.7 to 75.3) versus 53.3% (95% CI: 34.3 to 71.7).

**Conclusions** CARM is comparable with medical judgements in discriminating in-hospital mortality following emergency admission to an elderly care ward. CARM may have a promising role in supporting medical judgements in determining the patient's risk of death in hospital. Further evaluation of CARM in routine practice is required.

### Strengths and limitations of this study

► This study compares a novel computer-aided risk of mortality (CARM) score versus medical judgement in predicting the risk of in-hospital mortality.

► Consecutive emergency admissions to an elderly care ward in one hospital were assigned a risk of death at the first post-take ward round by consultant staff.

► The consultant medical staff used the same variables to assign a risk of death to the patient as the CARM (age, sex, National Early Warning Score and blood test results) but also had access to the clinical history, examination findings and any immediately available investigations such as ECGs.

► For a one-fourth of admissions with one or more blood test, missing CARM was similar to medical judgement with imputed blood test results.

## INTRODUCTION

Over the past few decades, numerous scoring systems have been developed to estimate the risk of mortality in hospital settings including intensive care medicine emergency medicine[1] and to a lesser extent general medical wards.[2] Despite the preponderance of scoring systems, systematic reviews[2] have highlighted a lack robust evaluation of risk scoring systems and only a few studies[3–5] have assessed their accuracy versus medical judgements in routine clinical settings. This is important because if the risk score is found not to perform well when compared with medical judgements, this would call into question the benefit of using the score in routine clinical practice. In a review of 12 studies in intensive care, Sinuff *et al*[6] found that physicians were better able to discriminate between

survivors and non-survivors than scoring systems in the first 24 hours of admission. However, one of their included studies[4] found that for patients at the extremes of risk of deterioration, clinicians outperformed scoring systems when assessing these patients but when assessing the 'in-between' group of patients, scoring systems were better than clinical judgement.[4]

We recently developed a computer-aided risk of mortality (CARM) score, which combines age, sex, vital signs (based on National Early Warning Score (NEWS)[7]) and seven blood test results for emergency medical admissions.[8] A key design feature of CARM is that it uses data which is already collected as part of the process of care and so places no additional data collection burden on clinicians. Furthermore, CARM is intended for computerised implementation and is not suited to pencil and paper methods because the underlying equation is not simple[8] as it involves 22 covariates with and without transformations and interaction effects. Nonetheless, it is important to note that CARM is intended to support, not displace, clinical judgement but the extent to which it can support the clinical decision-making process in practice remains unknown. So, as part of the ongoing evaluation of CARM, we set out to compare the performance of CARM versus medical judgements in estimating the risk of in-hospital mortality in consecutive emergency admissions to elderly care wards in one hospital over a 2-week period.

## METHODS
### Setting and data
Our cohort of elderly medical admissions is from York Hospital (managed by York Teaching Hospitals National Health service (NHS) Foundation Trust) which has approximately 700 beds. It has been exclusively using electronic NEWS scoring since 2013 as part of their in-house electronic patient record systems. Consecutive admissions to an elderly care medical admissions ward in this hospital were assigned a risk of death at the first post-take ward round by consultant medical staff over a 2-week period (05 February 2017 to 20 February 2017). The consultant medical staff used the same variables to assign a risk of death to the patient as the CARM (age, sex, NEWS and blood test results)[8] but also had access to the clinical history, examination findings and any immediately available investigations such as ECGs. Both CARM and medical judgements had access to the same physiological and pathological variables. The medical staff did not have access to the CARM score during the data collection exercise. For each admission, we obtained

the patient's age, sex (male/female), admission and discharge date and time, acute kidney injury (AKI) score, electronic NEWS (including its subcomponent vital signs data) and seven blood test results (albumin, creatinine, haemoglobin, potassium, sodium, urea and white cell count), although not all patients have all seven blood tests. To derive a CARM score for patients with missing blood test results, we imputed population-based age–sex median values. The reason for missing blood tests was that they were not ordered by the medical staff.

### Statistical analysis
The performance of CARM versus medical judgement was assessed by comparing risk estimates using boxplots. The discrimination of CARM and medical judgements was quantified by the area under the receiver-operating characteristic (ROC) curve or c-statistic.[9] In general, values less than 0.7 are considered to show poor discrimination, values of 0.7–0.8 can be described as reasonable, and values above 0.8 suggest good discrimination.[10] We compared the c-statistic for CARM and medical judgement using DeLong's test.[11]

We determined the sensitivity, specificity, positive predictive value (PPV) and negative predictive value (NPV), and positive and negative likelihood ratios for CARM and compared this with medical judgement scores using probability thresholds from a NEWS only model for NEWS scores from 1 to 5. The cut-off of NEWS at 5 is the recommended threshold for escalation of care.[12 13] We have also reported the geometric mean of sensitivity and specificity.[14]

All analyses were undertaken in STATA[15] and R[16] using rms[17] and pROC[18] packages.

### Ethical approval
This study received ethical approval from The Yorkshire & Humberside Leeds West Research Ethics Committee on 17 September 2015 (ref. 173753) with NHS management permissions received January 2016.

### Patient and public involvement
A workshop with a patient and service user group, linked to the University of Bradford, was involved at the start of this project to co-design the agenda for the patient and staff focus groups which were subsequently held at each hospital site. Patients were invited to attend the patient focus group through existing patient and public involvement groups. The criterion used for recruitment to these focus groups was any member of the public who had been a patient or carer in the last 5 years. The patient and

**Table 1** Pattern of missing blood test results in discharged alive/deceased elderly medical admissions

| Characteristic | Discharged alive (%) | Discharged deceased (%) | All (%) |
|---|---|---|---|
| Total emergency medical admissions | 279 | 130 | 409 |
| Complete blood test results recorded (%) | 202 (72.4) | 98 (75.4) | 300 (73.3) |
| At least one blood test result is not recorded (%) | 77 (27.6) | 32 (24.6) | 109 (26.7) |

**Table 2** Characteristics of all elderly medical admissions

| Characteristic | Discharged alive | Discharged deceased |
|---|---|---|
| n=409 | 279 | 130 |
| Male (%) | 123 (44.1) | 68 (52.3) |
| Mean CARM score (SD) | 0.07 (0.07) | 0.16 (0.16) |
| Mean medical judgement risk score (SD) | 0.12 (0.14) | 0.26 (0.25) |
| Mean NEWS (SD) | 2 (2.0) | 3.2 (3.2) |
| Alertness | | |
| Alert (%) | 278 (99.6) | 123 (94.6) |
| Pain (%) | 0 (0.0) | 3 (2.3) |
| Voice (%) | 1 (0.4) | 4 (3.1) |
| Unconscious (%) | 0 (0.0) | 0 (0.0) |
| AKI score | | |
| 0 (%) | 271 (97.1) | 122 (93.8) |
| 1 (%) | 5 (1.8) | 5 (3.8) |
| 2 (%) | 2 (0.7) | 2 (1.5) |
| 3 (%) | 1 (0.4) | 1 (0.8) |
| Oxygen supplementation (%) | 50 (17.9) | 42 (32.3) |
| Mean age (years) (SD) | 84.4 (5.5) | 86.7 (6.6) |
| Mean respiratory rate (breaths per minute) (SD) | 18.3 (2.9) | 19.1 (4.4) |
| Mean temperature (°C) (SD) | 36.5 (0.7) | 36.4 (0.8) |
| Mean systolic pressure (mm Hg) (SD) | 135.8 (25) | 124.1 (23.6) |
| Mean diastolic pressure (mm Hg) (SD) | 71 (13.8) | 68.2 (12.4) |
| Mean pulse rate (beats per minute) (SD) | 78.6 (16.4) | 81.6 (18.3) |
| Mean % oxygen saturation (SD) | 96.1 (2) | 95.5 (3.1) |
| Mean albumin (g/L) (SD) | | |
| No imputation (n=313) | 36.7 (4.3) | 33.6 (5.8) |
| With imputation (n=96)* | 36.8 (0.6) | 36.7 (1.0) |
| Mean creatinine (umol/L) (SD) | | |
| No imputation (n=391) | 103.3 (59.2) | 118.7 (75.3) |
| With imputation (n=18)* | 91.7 (10.8) | 88.7 (15.3) |
| Mean haemoglobin (g/L) (SD) | | |
| No imputation (n=391) | 123.3 (20.4) | 117.8 (17.7) |
| With imputation (n=18)* | 121.5 (4.4) | 116.5 (5.0) |
| Mean potassium (mmol/L) (SD) | | |
| No imputation (n=367) | 4.3 (0.5) | 4.4 (0.6) |
| With imputation (n=42)* | 4.3 (0.1) | 4.3 (0.1) |
| Mean sodium (mmol/L) (SD) | | |
| No imputation (n=383) | 136.1 (4.5) | 135.5 (5.7) |
| With imputation (n=26)* | 137.0 (0.4) | 136.8 (0.4) |
| Mean white cell count ($10^9$ cells/L) (SD) | | |
| No imputation (n=391) | 10.4 (6.4) | 11.8 (12.8) |
| With imputation (n=18)* | 9.2 (0.3) | 9.25 (0.2) |
| Mean urea (mmol/L) (SD) | | |
| No imputation (n=391) | 9.2 (5.3) | 12.3 (8.9) |
| With imputation (n=18)* | 8.3 (0.8) | 7.9 (1.4) |

*Imputed blood test results using age-specific and sex-specific population median values.
AKI, acute kidney injury; CARM, computer-aided risk of mortality; NEWS, National Early Warning Score.

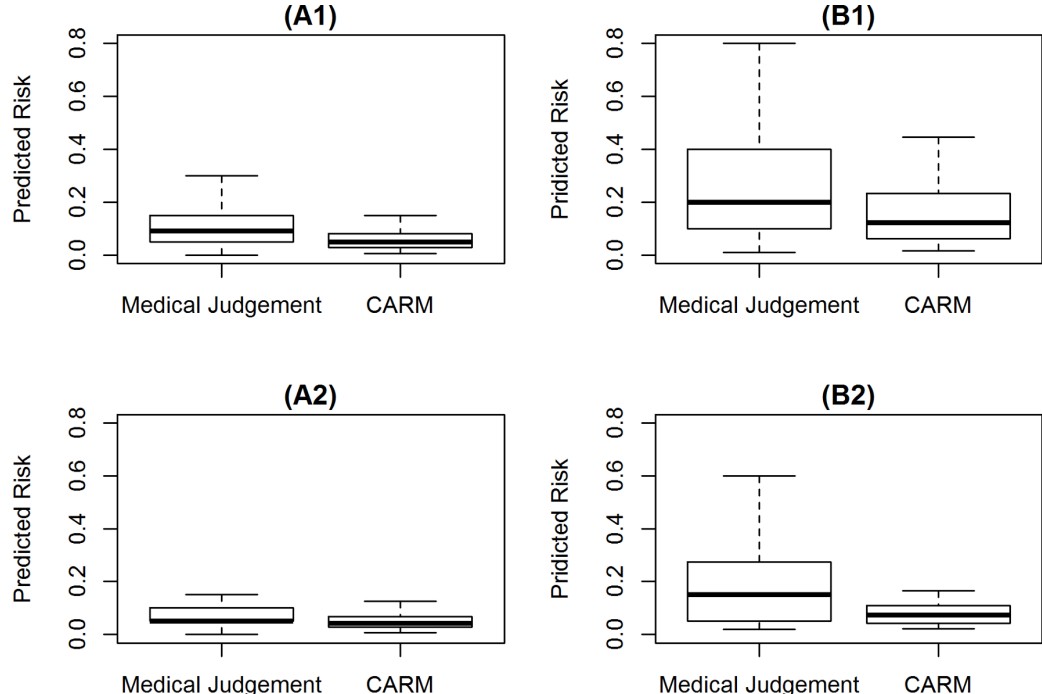

**Figure 1** Comparison of medical judgement versus CARM in predicting risk of mortality for patients who (A) discharged alive and (B) discharged deceased. (A1/B1) Complete blood test results (n=300); (A2/B2) at least one blood test result is imputed (n=109). CARM, computer-aided risk of mortality.

public voice continued to be included throughout the project with three patient representatives invited to sit on the project steering group. Participants will be informed of the results of this study through the patient and public involvement leads at each hospital site and the project team have met with the Bradford Patient and Service User Group to discuss the results.

## RESULTS
### Cohort description
The study involved 409 emergency medical admissions to the elderly care wards in York Hospital. Of these, 300 (73.3%) had a full set of blood test and 109 (26.7%) had at least one blood test result missing (table 1). The most frequent missing blood test was albumin (n=96).

The in-hospital mortality was 31.8% (130/409). The age, sex, NEWS and blood test results profile are shown in table 2. Compared with patients discharged alive, deceased patients were aged older, with lower albumin, haemoglobin and sodium values, and higher creatinine, potassium, white cell count and urea values. NEWS was higher in deceased patients compared with patients discharged alive, as were respiratory rate and pulse rate values. The temperature, blood pressure and oxygen saturation values were lower in deceased patients. Where blood test results were missing, we imputed the age– sex population median value which appeared to give more reasonable values for patients discharged alive than those who died (see imputed values in table 2 comparing imputed values with observed values). For example, the observed mean (n=313) for albumin is 36.7 for survivors

versus 33.6 for non-survivors. However, the imputed means for albumin (n=96) were 36.8 for survivors and 36.7 for non-survivors.

### Comparison of CARM versus medical judgement
Figure 1 shows the estimated risk of in-hospital mortality using CARM versus medical judgements for patients who discharged alive and deceased. The mean estimated risk of in-hospital mortality for patients discharged alive was lower with CARM (0.07 SD=0.07) versus medical judgements (0.12 SD=0.14). Likewise, for decreased patients, the risk estimates from CARM (0.16 SD=0.16) were lower than estimates from medical judgements (0.26 SD=0.25) (see table 2).

Figure 2 shows the ROC curve. The area under the ROC curve (c-statistic) was higher for CARM 0.75 (95% CI: 0.69 to 0.81) than for medical judgement 0.72 (95% CI: 0.66 to 0.78) and were not statistically significant (p value=0.28). The area under the ROC curve was similar for admissions with at least one blood test result missing (see table 3).

Table 4 shows the sensitivity, specificity, PPV and NPV for a selected range of NEWS values. For patients with complete blood test results (n=300), NEWS at 5 (the recommended escalation threshold), which is equivalent to a 10% risk of in-hospital mortality, medical judgement had a higher sensitivity 59.2% (95% CI: 48.8 to 69.0) versus 58.2% (95% CI: 47.8 to 68.1), lower specificity 70.3% (95% CI: 63.5 to 76.5) versus 82.7% (95% CI: 76.7 to 87.6), lower PPVs 49.2% (95% CI: 39.8 to 58.5) versus 62.0% (95% CI: 51.2 to 71.9) and a lower positive likelihood ratio (2 vs 3.4) than the CARM score.

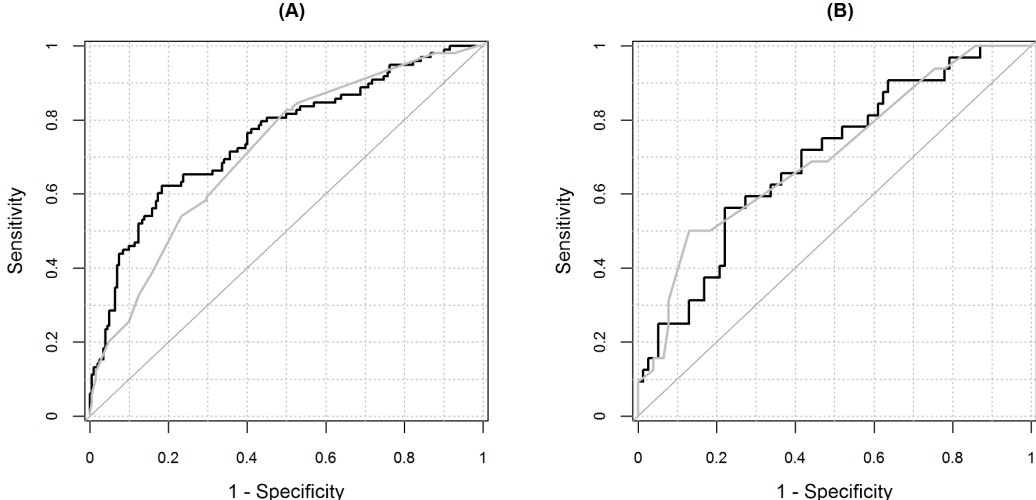

**Figure 2** Receiver operating characteristic curve for CARM and medical judgements with/without imputed blood test results. (A) Complete blood test results (n=300); (B) at least one blood test result is imputed (n=109). Black line is for CARM and grey line is for medical judgement. CARM, computer-aided risk of mortality.

For patients with at least one imputed blood test result (n=109), at a NEWS of 5 medical judgement had a higher sensitivity 50.0% (95% CI: 31.9 to 68.1) versus 25.0% (95% CI: 11.5 to 43.4), lower specificity 81.8% (95% CI: 71.4 to 89.7) versus 89.6% (95% CI: 80.6 to 95.4), higher PPVs 53.3% (95% CI: 34.3 to 71.7) versus 50.0% (95% CI: 24.7 to 75.3) and higher positive likelihood ratios (2.8 vs 2.4).

## DISCUSSION

In this study, we assessed the accuracy of CARM versus medical judgements in consecutive emergency admissions to the elderly care ward over a 2-week period. We found for patients with complete blood test results, the c-statistic for CARM was 0.75 versus 0.72 for medical judgements (p=0.28). For patients with at least one missing blood test result, the c-statistics were lower but still similar (medical judgements 0.70 vs CARM 0.70). At a 10% mortality risk, the PPV for CARM was higher than medical judgements in patients with complete blood test results (62.0% vs 49.2%) but not when blood test results were missing (50.0% vs 53.3%).

Overall, when comparing CARM with medical judgements, no significant differences in area under the curve were found. These findings are remarkable because, unlike medical judgements, CARM relies exclusively on routinely collected data based primarily on the patients' age, vital signs and blood test results without having any

disease labels or clinical history. Furthermore, where blood tests are being imputed, CARS and medical judgements are less able to discriminate mortality. While this is to be expected for CARM because we use a population median imputation strategy, which is biased towards survivors, the reasons for lower c-statistics for medical judgements are less clear. It would suggest that these patients (with one or more missing blood test results) are more challenging to assess for the medical staff although the underlying reasons are not clear.

Our findings are in line with other studies, which also found no significant differences between ROC curves for Acute Physiology and Chronic Health Evaluation (APACHEII) score and clinical staff.[19] However, a study reported that the clinical assessment had an overall accuracy of 95.2% versus 90.9% for APACHE2.[3] Other studies have also failed to show an advantage for the APACHE2 model when compared with medical judgements by the clinicians.[4 5 20] Another study found that physicians were significantly better in predicting outcome in a medical intensive care unit than APACHE.[21] One study concluded that physicians' clinical judgement could differ from scoring systems enough to account for large differences in expected outcomes.[20]

It is important to note that we have designed CARM to support the medical decision-making process, not replace it, without placing any additional data collection burden on staff. The CARM risk prediction can also be made

**Table 3** Comparing discrimination of medical judgement versus CARM in predicting the risk of in-hospital mortality

| Imputation | Medical judgement AUC (95% CI) | CARM AUC (95% CI) | P value |
|---|---|---|---|
| Complete blood test results (n=300) | 0.72 (0.66 to 0.78) | 0.75 (0.69 to 0.81) | 0.28 |
| At least one blood test result is imputed (n=109) | 0.70 (0.60 to 0.81) | 0.70 (0.59 to 0.80) | 0.86 |

AUC, area under the curve; CARM, computer-aided risk of mortality.

**Table 4** Performance of CARM versus medical judgement with/without imputation in predicting the risk in-hospital mortality at NEWS thresholds (1, 2, 3, 4 and 5)

| | | Medical judgement | | | | | | | | CARM | | | | | | | |
|---|---|---|---|---|---|---|---|---|---|---|---|---|---|---|---|---|---|
| NEWS | Predicted risk at NEWS thresholds | N | Sensitivity% | Specificity% | PPV | NPV | LR+ | LR− | GM% | N | Sensitivity% | Specificity% | PPV | NPV | LR+ | LR− | GM% |
| Complete blood test results n=300 | | | | | | | | | | | | | | | | | |
| 1 | 0.03 | 275 | 98.0 (92.8 to 99.8) | 11.4 (7.4 to 16.6) | 34.9 (29.3 to 40.9) | 92.0 (74 to 99) | 1.1 (1.0 to 1.2) | 0.2 (0.0 to 0.7) | 33.4 | 239 | 90.8 (83.3 to 95.7) | 25.7 (19.9 to 32.3) | 37.2 (31.1 to 43.7) | 85.2 (73.8 to 93) | 1.2 (1.1 to 1.4) | 0.4 (0.2 to 0.7) | 48.4 |
| 2 | 0.04 | 273 | 98.0 (92.8 to 99.8) | 12.4 (8.2 to 17.7) | 35.2 (29.5 to 41.1) | 92.6 (75.7 to 99.1) | 1.1 (1.1 to 1.2) | 0.2 (0.0 to 0.7) | 34.8 | 205 | 84.7 (76 to 91.2) | 39.6 (32.8 to 46.7) | 40.5 (33.7 to 47.5) | 84.2 (75.3 to 90.9) | 1.4 (1.2 to 1.6) | 0.4 (0.2 to 0.6) | 57.9 |
| 3 | 0.05 | 190 | 84.7 (76 to 91.2) | 47.0 (40.0 to 54.2) | 43.7 (36.5 to 51.1) | 86.4 (78.5 to 92.2) | 1.6 (1.4 to 1.9) | 0.3 (0.2 to 0.5) | 63.1 | 168 | 79.6 (70.3 to 87.1) | 55.4 (48.3 to 62.4) | 46.4 (38.7 to 54.3) | 84.8 (77.6 to 90.5) | 1.8 (1.5 to 2.1) | 0.4 (0.2 to 0.6) | 66.4 |
| 4 | 0.08 | 186 | 83.7 (74.8 to 90.4) | 48.5 (41.4 to 55.6) | 44.1 (36.8 to 51.5) | 86.0 (78.2 to 91.8) | 1.6 (1.4 to 1.9) | 0.3 (0.2 to 0.5) | 63.7 | 126 | 65.3 (55.0 to 74.6) | 69.3 (62.4 to 75.6) | 50.8 (41.7 to 59.8) | 80.5 (73.8 to 86.1) | 2.1 (1.7 to 2.7) | 0.5 (0.4 to 0.7) | 67.3 |
| 5 | 0.10 | 118 | 59.2 (48.8 to 69.0) | 70.3 (63.5 to 76.5) | 49.2 (39.8 to 58.5) | 78.0 (71.3 to 83.8) | 2.0 (1.5 to 2.6) | 0.6 (0.5 to 0.7) | 64.5 | 92 | 58.2 (47.8 to 68.1) | 82.7 (76.7 to 87.6) | 62.0 (51.2 to 71.9) | 80.3 (74.2 to 85.5) | 3.4 (2.4 to 4.7) | 0.5 (0.4 to 0.6) | 69.3 |
| At least one blood test result is imputed n=109 | | | | | | | | | | | | | | | | | |
| 1 | 0.03 | 89 | 93.8 (79.2 to 99.2) | 23.4 (14.5 to 34.4) | 33.7 (24.0 to 44.5) | 90.0 (68.3 to 98.8) | 1.2 (1.1 to 1.4) | 0.3 (0.1 to 1.1) | 46.8 | 83 | 90.6 (75.0 to 98.0) | 29.9 (20.0 to 41.4) | 34.9 (24.8 to 46.2) | 88.5 (69.8 to 97.6) | 1.3 (1.1 to 1.6) | 0.3 (0.1 to 1.0) | 52.0 |
| 2 | 0.04 | 88 | 93.8 (79.2 to 99.2) | 24.7 (15.6 to 35.8) | 34.1 (24.3 to 45) | 90.5 (69.6 to 98.8) | 1.2 (1.1 to 1.5) | 0.3 (0.1 to 1.0) | 48.1 | 63 | 75.0 (56.6 to 88.5) | 49.4 (37.8 to 61.0) | 38.1 (26.1 to 51.2) | 82.6 (68.6 to 92.2) | 1.5 (1.1 to 2.0) | 0.5 (0.3 to 1.0) | 60.8 |
| 3 | 0.05 | 59 | 68.8 (50.0 to 83.9) | 51.9 (40.3 to 63.5) | 37.3 (25.0 to 50.9) | 80.0 (66.3 to 90.0) | 1.4 (1.0 to 2.0) | 0.6 (0.3 to 1.0) | 59.8 | 47 | 62.5 (43.7 to 78.9) | 64.9 (53.2 to 75.5) | 42.6 (28.3 to 57.8) | 80.6 (68.6 to 89.6) | 1.8 (1.2 to 2.7) | 0.6 (0.4 to 0.9) | 63.7 |
| 4 | 0.08 | 59 | 68.8 (50.0 to 83.9) | 51.9 (40.3 to 63.5) | 37.3 (25.0 to 50.9) | 80.0 (66.3 to 90.0) | 1.4 (1.0 to 2.0) | 0.6 (0.3 to 1.0) | 59.8 | 30 | 40.6 (23.7 to 59.4) | 77.9 (67.0 to 86.6) | 43.3 (25.5 to 62.6) | 75.9 (65 to 84.9) | 1.8 (1.0 to 3.3) | 0.8 (0.6 to 1.0) | 56.3 |
| 5 | 0.10 | 30 | 50.0 (31.9 to 68.1) | 81.8 (71.4 to 89.7) | 53.3 (34.3 to 71.7) | 79.7 (69.2 to 88.0) | 2.8 (1.5 to 4.9) | 0.6 (0.4 to 0.9) | 64.0 | 16 | 25.0 (11.5 to 43.4) | 89.6 (80.6 to 95.4) | 50.0 (24.7 to 75.3) | 74.2 (64.1 to 82.7) | 2.4 (1.0 to 5.9) | 0.8 (0.7 to 1.0) | 47.3 |

CARM, computer-aided risk of mortality; GM, geometric mean; LR+, positive likelihood ratio; LR−, negative likelihood ratio; NEWS, National Early Warning Score; NPV, negative predictive value; PPV, positive predictive value.

available as soon as the physiological observations and blood test results are available and prior to the consultant review which may be of assistance to more junior staff. CARM was developed using all adult non-elective medical and elderly care admissions to in one hospital and externally validated in another hospital.[8]

The overall mortality was 5% in the study population in which the CARM risk predictor was developed. The overall mortality in this patient cohort is high and it is worth noting that patients had already been streamed (selected) as requiring in-patient admission as direct admission from GP or via the emergency department. Thus, the pre-test probability of mortality is different to original study population; yet, the CARM risk predictor still performs reasonably well in this population.

Our study has several limitations. This study provides a snapshot of the use of CARM in a hospital over a short period and the extent to which our findings generalise to patients over a longer time period and to other wards and hospitals require further study. Although CARM is designed to be automated, we note that for 26% of patients were unable to derive the CARM score because of no or incomplete blood test results and the most frequent missing blood test result was albumin. Although we adopted a median imputation strategy, the extent to which this is acceptable in routine clinical practice remains unknown especially as this imputation strategy is biased towards survivors and so will underestimate the true risk of dying for those who are likely to die. So further study is required to understand the issue of missing blood test results and how to address it in routine clinical practice. One possibility is that there may be an unintended increase in the use of blood test results in patients where blood test would not ordinarily be undertaken to simply provide a CARM score. Crucially, how the medical decision-making process is modified by the availably or CARM and the extent to which it enhances situational awareness and subsequently enhances the quality of care without adverse unintended consequences remains to be seen.

## CONCLUSIONS

CARM is comparable with medical judgements in predicting in-hospital mortality following emergency admission to an elderly care ward. CARM may have a promising role in supporting medical judgements in determining the patient's risk of death in hospital. Further evaluation of CARM in routine practice is required.

**Author affiliations**
[1]Faculty of Health Studies, University of Bradford, Bradford, UK
[2]School of Clinical Therapies, University College Cork National University of Ireland, Cork, Ireland
[3]Renal Unit, York Teaching Hospital NHS Foundation Trust, York, UK
[4]York Teaching Hospital NHS Foundation Trust, York, UK
[5]NHS Midlands and Lancashire Commissioning Support Unit, West Bromwich, UK

**Contributors** MM and DR had the original idea for this work. MF undertook the statistical analyses with guidance from AS and MAM. DR gave a clinical perspective. MF and BK wrote the first draft of this paper. SI, RD, DH, AC, JA, RH,

SK, GM, KG and MH contributed to data collection and all authors subsequently assisted in redrafting and have approved the final version. MM will act as guarantor.

**Funding** This research was supported by the Health Foundation. The Health Foundation is an independent charity working to improve the quality of health care in the UK. This research was supported by the National Institute for Health Research (NIHR) Yorkshire and Humberside Patient Safety Translational Research Centre (NIHR YHPSTRC).

**Disclaimer** The views expressed in this article are those of the author(s) and not necessarily those of the NHS, the NIHR, or the Department of Health and Social Care.

**Competing interests** None declared.

**Patient consent for publication** Not required.

**Provenance and peer review** Not commissioned; externally peer reviewed.

**Data sharing statement** Our data sharing agreement with the York hospital does not permit us to share this data with other parties. Nonetheless, if anyone is interested in the data, then they should contact the R&D offices at York hospital in the first instance.

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
