## [Reviewer comments · BMJ Open]

ARTICLE DETAILS

TITLE (PROVISIONAL)	A prospective study of consecutive emergency medical admissions to compare a novel automated computer aided mortality risk score and clinical judgement of patient mortality risk.
AUTHORS	Faisal, Muhammad; Khatoon, Binish; Scally, Andy; Richardson, Donald; Irwin, Sally; Davidson, Rachel; Heseltine, David; Corlett, Alison; Ali, Javed; Hampson, Rebecca; Kesavan, Sandeep; McGonigal, Gerry; Goodman, Karen; Harkness, Michael; Mohammed, Mohammed

VERSION 1 - REVIEW

REVIEWER	Candice Downey University of Leeds, United Kingdom
REVIEW RETURNED	29-Nov-2018

GENERAL COMMENTS	Introduction: There is limited background information provided for the reader. Some of this is provided in the Discussion but would be better placed earlier in the paper. Please indicate in the introduction the benefits of mortality risk prediction in your chosen population. These should be summarised as benefits to patients, carers, ward staff, hospitals and the Health Service as a whole. Please briefly explain why scores are required in addition to clinical judgment. How accurate is clinical judgment? Please briefly explain why the new mortality score (CARM) has been developed given the "numerous scoring systems" that already exist. Methods: Were the medical staff blinded to the nature of the study? You have stated in the Introduction that all of the data items used in CARM are routinely collected. In the Methods, you state that albumin is not routinely included in the list of routine blood tests at York hospital. Please clarify. The explanation of ROC curves and AUCs is probably unnecessary. Simply refer interested readers to a relevant paper for further information. Results: Page 10 - please explain what is 'not statistically significant' for clarity. Please reference the claim that a NEWS of 5 is associated with 10% mortality risk.
--

	Discussion: Given that the statistical analysis did not show a significant difference between CARM and clinical judgment, is it correct to say that CARM compares favourably? You could certainly say the two methods were comparable, but the data does not support one over the other. Please also clarify if the CARM does in fact use routinely collected data, when albumin is not routinely reported in York Hospital. Could a sensitivity analysis be performed using the records from patients who did not undergo blood tests, to ascertain if the issues highlighted in paragraph 2 are valid? Please clarify which exact populations you recommend the CARM should be used in. Minor points: Throughout: Please use judgment/judgement consistently throughout the manuscript. Abstract: Please delete the (CARM) in the second sentence of the Results section. Article Summary: The third sentence needs a full stop. Introduction: Typos include 'sever' for 'severe' (second paragraph) Methods: third line, delete 'at least' or explain Explain AKI score 'We excluded records where blood test results were not undertaken at all' should read 'We excluded records where bloods tests had not been undertaken.' Please avoid starting a sentence with 'However' 'We further determined the sensitivity, specificity, positive and negative predictive values, positive and negative likelihood ratios for CARM...' should read 'We further determined the sensitivity, specificity, positive and negative predictive values, and positive and negative likelihood ratios for CARM...' The Patient and Public Involvement section: correct 'these focus group' and consider use of capitals in 'Bradford Patient and service user group' Please refer to York Hospital in a consistent manner throughout the manuscript. Explain R&D Results: Comparison of CARM versus Medical Judgement section - the first sentence is nonsensical. Suggest delete 'that'. The second sentence is redundant. Discussion: Fourth paragraph, second sentence, please amend study/studies?
--	--

REVIEWER	Jose Antonio Sanz Universidad Publica de Navarra, Spain
REVIEW RETURNED	19-Dec-2018

GENERAL COMMENTS	In this paper authors propose the application of automated method named CARM to predict the patient's risk of mortality. I have some comments that need to be addressed before accepting the manuscript:  1. The CARM method should be briefly described to make the paper self contained. In the same sense, it would be interesting how the medical judgment risk is computed.
--

	2. It would be interesting to apply the references methods (APACHE2 and SAPS) over the data of this study so that they could be objectively compared instead of mentioning other studies as make in the discussion section. 3. In the cohort description the in-hospital mortality numbers do not coincide between the text and the table. Please check it. 4. I wonder which the threshold used to determine whether a patient dies or not is. I understand that to compute the AUC is not needed but it is needed to predict new patients. 5. In the last paragraph of page 10 authors shows sensitivity and other metrics based on NEWS values, which determine the risk percentage. Is this percentage the threshold used to determine whether a patient dies or not? Please clarify this fact. Moreover, a well know metric used in imbalanced domains is the geometric mean between the sensitivity and the specificity (I attach a reference where you can find it below), since it summarizes both metrics and shows the balance of the method. Therefore, I suggest including it to make the analysis more complete. <pre>@article{SANZ20171, title = "A new survival status prediction system for severe trauma patients based on a multiple classifier system", journal = "Computer Methods and Programs in Biomedicine", volume = "142", pages = "1 - 8", year = "2017", issn = "0169-2607", doi = "https://doi.org/10.1016/j.cmpb.2017.02.011", }</pre>
--	--

REVIEWER	Alai Tan The Ohio State University USA
REVIEW RETURNED	21-Dec-2018

GENERAL COMMENTS	The study compared the performance of CARM vs. medical judgement in predicting risk of in-hospital mortality. The risk prediction is important for clinical care. However, the study results could not support the authors' conclusion on the incremental benefit of CARM other than quick computation. Abstract: The conclusion that "CARM compares favorably with medical judgements in routine clinical care" is not supported by the data: 1) Although AUC was higher for CARM (vs. medical judgement), the difference was small (0.75 vs 0.72) and not statistically significant indicated by the 95% Cis; 2) Practically, medical judgement appears to have advantages since CARM were not applicable to 12% of the patients due to missing NEWS and blood test results (page 4). Main Text Page 6, lines 15-17. Need details on what tools or information the consultant medical staff used to assign a risk of death to the patient. It is likely that age, sex, severity illness, etc. were included in the consideration. If both methods (CARM vs. medical judgements) were based on same sets of information, then the advantage of CARM will be the automation of the computation, resulting in more rapid and efficient prediction than the labor
--

	intensive medical judgement rather than the improvement in accuracy. Page 6, lines 35-37. It is not clear why data on albumin can be imputed even though it was not a routine blood test at York Hospital, while other routine blood tests were not imputable if missing. What are the pattern of the missing data? Did the imputation impact the results? Pages 8-9, table 2. Need test statistics on the comparison of the characteristics between those deceased and those alive at discharge. Also it will be helpful to organize the characteristics into groups: those solely used for medical judgement, those solely used for CARM, and those included by both methods. This information will be helpful to determine whether parameters included in CARM can be further trimmed down to have a more parsimonious model. Results section did not have details of the CARM model and how the prediction achieved, based on coefficients or based on assigned weights for the predictors? What are the significant predictors and what are the non-significant predictors? The model fit statistics? Especially those adjusted for the number of predictors (e.g., AIC and BIC). Any internal/external validation was conducted? Pages 10, lines 8-12. Did the authors conduct any calibration since “The predicted risk is systematically lower using CARM than for medical judgement for both patients who were discharged alive and deceased”? This is another evidence that did not support the authors’ conclusion that “CARM compares favorably with medical judgement” in addition to the two points mentioned earlier. Page 12, lines 5-12. Again, the conclusion was not supported by the study findings. Although “CARM has a comparable discrimination, and higher PPV and positive likelihood ratios”, it has lower sensitivity and systematically underestimate the risk of death. Some patients who actually have higher risk of death could potentially miss the opportunity of receiving escalated care based on CARM prediction. In addition, it was not applicable to 12% of the patients due to missing data. Other than the quick computation, the incremental benefit of CARM to medical judgement is not convincing.
--	--

VERSION 1 – AUTHOR RESPONSE

Reviewer(s)' Comments to Author:

Reviewer: 1

Reviewer Name: Candice Downey

Institution and Country: University of Leeds, United Kingdom

Please state any competing interests or state 'None declared': None declared

Please leave your comments for the authors below

1. Introduction: There is limited background information provided for the reader. Some of this is provided in the Discussion but would be better placed earlier in the paper.

Response: We have now revised the introduction and moved some sections to the discussion as suggested.

2. Please indicate in the introduction the benefits of mortality risk prediction in your chosen population. These should be summarised as benefits to patients, carers, ward staff, hospitals and the Health Service as a whole.

Response: We have now done this in the introduction. Some of the key benefits of using mortality risk prediction scores is the recognition of deterioration in patients, reduction of unnecessary harm in hospitals, reduced hospital costs and prompt delivery of appropriate medical interventions and escalation of care.

3. Please briefly explain why scores are required in addition to clinical judgment. How accurate is clinical judgment?

Response: We have provided further explanation in the introduction. In a critical care setting, for patients at the extremes of risk of deterioration, clinicians outperform scoring systems when assessing these groups of people. Similarly, patients who are doing very well or very poorly are easily identified also, but when assessing the in-between group, scoring systems are better than clinical experience. Clinical judgement is comparable with the medical judgement in terms of AUC.

4. Please briefly explain why the new mortality score (CARM) has been developed given the "numerous scoring systems" that already exist.

Response: This is now included in the introduction. An important feature of CARM is that it is not designed for paper based systems and do not place any additional burden of data collection and/or calculation on the clinicians because it relies on data which are (a) routinely collected as part of the process of care, (b) already stored in the patient's electronic health record and (c) accessible in real-time thus offering the prospects of real-time risk predictions without hindering clinical workflows. Furthermore, CARM is not intended to replace clinical judgement but be used alongside clinical judgement.

Methods: Were the medical staff blinded to the nature of the study?

Response: No – they were not blinded to the nature of the study but the clinicians made their estimations prospectively at the time of the first consultant ward round after admission without access to the CARM risk.

You have stated in the Introduction that all of the data items used in CARM are routinely collected. In the Methods, you state that albumin is not routinely included in the list of routine blood tests at York hospital. Please clarify.

Response: We have now clarified that Albumin is included in the list of routine blood tests, however it is not always ordered by clinicians.

The explanation of ROC curves and AUCs is probably unnecessary. Simply refer interested readers to a relevant paper for further information.

Response: We have now removed the explanation of ROC curves as suggested.

Results: Page 10 - please explain what is 'not statistically significant' for clarity. Please reference the claim that a NEWS of 5 is associated with 10% mortality risk.

Response: We have now clarified that 'not statistically significant' means medical judgement and CARM are comparable in terms of discriminating between discharged alive or deceased patients. We have provided the predicted risk of in-hospital mortality at various NEWS thresholds (1,2,3,4,5) in table 4. The NEWS at 5 is equivalent to 10% predicted risk of mortality in our data set. This would change with different patient subgroups.

Discussion: Given that the statistical analysis did not show a significant difference between CARM and clinical judgment, is it correct to say that CARM compares favourably? You could certainly say the two methods were comparable, but the data does not support one over the other.

Please also clarify if the CARM does in fact use routinely collected data, when albumin is not routinely reported in York Hospital.

Response: We have now made it clearer in the manuscript that CARM is comparable to Medical Judgement. We have now clarified that Albumin is routinely available blood test, but it is not always ordered by clinicians in York Hospital.

Could a sensitivity analysis be performed using the records from patients who did not undergo blood tests, to ascertain if the issues highlighted in paragraph 2 are valid?

Response: We have now performed the sensitivity analysis and provided the results with/without imputation.

Please clarify which exact populations you recommend the CARM should be used in.

Response: All adult (age \geq 16 years) emergency medical admission with complete blood test results who are being cared for in the ward (not ICU – which has its own risk scoring system) The use of CARM for similar patients without full blood test results is not recommended until further research.

Minor points:

Throughout: Please use judgment/judgement consistently throughout the manuscript.

Abstract: Please delete the (CARM) in the second sentence of the Results section.

Article Summary: The third sentence needs a full stop.

Introduction: Typos include 'sever' for 'severe' (second paragraph)

Methods: third line, delete 'at least' or explain Explain AKI score 'We excluded records where blood test results were not undertaken at all' should read 'We excluded records where bloods tests had not been undertaken.'

Please avoid starting a sentence with 'However'

'We further determined the sensitivity, specificity, positive and negative predictive values, positive and negative likelihood ratios for CARM...' should read 'We further determined the sensitivity, specificity, positive and negative predictive values, and positive and negative likelihood ratios for CARM...'

The Patient and Public Involvement section: correct 'these focus group' and consider use of capitals in 'Bradford Patient and service user group'

Please refer to York Hospital in a consistent manner throughout the manuscript.

Explain R&D

Results: Comparison of CARM versus Medical Judgement section - the first sentence is nonsensical. Suggest delete 'that'. The second sentence is redundant.

Discussion: Fourth paragraph, second sentence, please amend study/studies?

Response: We have now made changes as suggested.

Reviewer: 2

Reviewer Name: Jose Antonio Sanz

Institution and Country: Universidad Publica de Navarra, Spain

Please state any competing interests or state 'None declared': None declared

Please leave your comments for the authors below In this paper authors propose the application of automated method named CARM to predict the patient's risk of mortality. I have some comments that need to be addressed before accepting the manuscript:

1. The CARM method should be briefly described to make the paper self contained. In the same sense, it would be interesting how the medical judgment risk is computed.

Response: We have now briefly described the CARM method as suggested.

2. It would be interesting to apply the references methods (APACHE2 and SAPS) over the data of this study so that they could be objectively compared instead of mentioning other studies as make in the discussion section.

Response: Unfortunately, the data about references methods (APACHE2 and SAPS) is not available. Therefore, we are unable to compare CARM with them. Furthermore, CARM is intended for general medical wards whereas APACHE2 is intended for use in ICU.

3. In the cohort description the in-hospital mortality numbers do not coincide between the text and the table. Please check it.

Response: We have now corrected it. Thank you.

4. I wonder which the threshold used to determine whether a patient dies or not is. I understand that to compute the AUC is not needed but it is needed to predict new patients.

Response: We have already discussed this issue in our CARM paper. We found 0.08 to be a reasonable threshold in terms of accuracy. See Table S9

<https://bmjopen.bmj.com/content/bmjopen/8/12/e022939/DC1/embed/inline-supplementary-material-1.pdf>

5. In the last paragraph of page 10 authors shows sensitivity and other metrics based on NEWS values, which determine the risk percentage. Is this percentage the threshold used to determine whether a patient dies or not? Please clarify this fact. Moreover, a well know metric used in imbalanced domains is the geometric mean between the sensitivity and the specificity (I attach a reference where you can find it below), since it summarizes both metrics and shows the balance of the method. Therefore, I suggest including it to make the analysis more complete.

@article{SANZ2017,

title = "A new survival status prediction system for severe trauma patients based on a multiple classifier system", journal = "Computer Methods and Programs in Biomedicine", volume = "142", pages = "1 - 8", year = "2017", issn = "0169-2607", doi = "https://doi.org/10.1016/j.cmpb.2017.02.011",

}

Response: We have now reported the proposed measure.

Reviewer: 3

Reviewer Name: Alai Tan

Institution and Country: The Ohio State University, USA

Please state any competing interests or state 'None declared': None

Please leave your comments for the authors below The study compared the performance of CARM vs. medical judgement in predicting risk of in-hospital mortality. The risk prediction is important for clinical care. However, the study results could not support the authors' conclusion on the incremental benefit of CARM other than quick computation.

Response: We have imputed population median for missing blood test results matched on age and sex. CARM can be used for all the patients, however the performance of CARM is similar to medical judgement in imputed blood tests result data. From another study, we found that Medical Judgement is mostly based on clinical history and this has been not considered in CARM. We now made it clearer that CARM is comparable to Medical Judgement.

Abstract:

The conclusion that “CARM compares favorably with medical judgements in routine clinical care” is not supported by the data: 1) Although AUC was higher for CARM (vs. medical judgement), the difference was small (0.75 vs 0.72) and not statistically significant indicated by the 95% Cis; 2) Practically, medical judgement appears to have advantages since CARM were not applicable to 12% of the patients due to missing NEWS and blood test results (page 4).

Response: We now made it clearer that CARM is comparable to Medical Judgement.

Main Text

Page 6, lines 15-17. Need details on what tools or information the consultant medical staff used to assign a risk of death to the patient. It is likely that age, sex, severity illness, etc. were included in the consideration. If both methods (CARM vs. medical judgements) were based on same sets of information, then the advantage of CARM will be the automation of the computation, resulting in more rapid and efficient prediction than the labor intensive medical judgement rather than the improvement in accuracy.

Response: We have imputed population median for missing blood test results matched on age and sex. So, CARM can be used for all the patients, however the performance of CARM is similar to medical judgement in imputed blood tests result data. From another study, we found that Medical Judgement is mostly based on clinical history and this has been not considered in CARM. We now made it clearer that CARM is comparable to Medical Judgement.

Page 6, lines 35-37. It is not clear why data on albumin can be imputed even though it was not a routine blood test at York Hospital, while other routine blood tests were not imputable if missing. What are the pattern of the missing data? Did the imputation impact the results?

Response: We have now imputed Albumin and other routine blood tests and compared the CARM with Medical Judgement in terms of AUC and sensitivity analysis.

Pages 8-9, table 2. Need test statistics on the comparison of the characteristics between those deceased and those alive at discharge. Also it will be helpful to organize the characteristics into groups: those solely used for medical judgement, those solely used for CARM, and those included by both methods. This information will be helpful to determine whether parameters included in CARM can be further trimmed down to have a more parsimonious model.

Response: Our aim is to combined routinely collected blood test results and vital signs. We are not aiming for parsimonious model.

Results section did not have details of the CARM model and how the prediction achieved, based on coefficients or based on assigned weights for the predictors? What are the significant predictors and what are the non-significant predictors? The model fit statistics? Especially those adjusted for the number of predictors (e.g., AIC and BIC). Any internal/external validation was conducted?

Response: Yes –CARM model has been externally validated. Our aim is to combined routinely collected blood test results and vital signs. We are not looking for parsimonious model. We have reported AUC, Brier score, and discrimination slope. Please see the original paper at <https://bmjopen.bmj.com/content/bmjopen/8/12/e022939/DC1/embed/inline-supplementary-material-1.pdf>

Pages 10, lines 8-12. Did the authors conduct any calibration since “The predicted risk is systematically lower using CARM than for medical judgement for both patients who were discharged alive and deceased”? This is another evidence that did not support the authors’ conclusion that “CARM compares favorably with medical judgement” in addition to the two points mentioned earlier.

Response: We have noted this as limitation in the discussion section. Moreover, we now made it clearer that CARM is comparable to Medical Judgement.

Page 12, lines 5-12. Again, the conclusion was not supported by the study findings. Although “CARM has a comparable discrimination, and higher PPV and positive likelihood ratios”, it has lower sensitivity and systematically underestimate the risk of death. Some patients who actually have higher risk of death could potentially miss the opportunity of receiving escalated care based on CARM prediction. In addition, it was not applicable to 12% of the patients due to missing data. Other than the quick computation, the incremental benefit of CARM to medical judgement is not convincing.

Response: We have imputed population median for missing blood test results matched on age and sex. So, CARM can be used for all the patients, however the performance of CARM is similar to medical judgement in imputed blood tests result data. From another study, we found that Medical Judgement is mostly based on clinical history and this has been not considered in CARM. We now made it clearer that CARM is comparable to Medical Judgement. Furthermore, CARM is not intended to replace clinical judgement but be used alongside clinical judgement.

VERSION 2 – REVIEW

REVIEWER	Candice Downey University of Leeds, United Kingdom
REVIEW RETURNED	09-Apr-2019

GENERAL COMMENTS	This paper is much improved given the more balanced conclusions drawn by the authors. The work adds value to the literature in the field. Recommend accept.
---

REVIEWER	Jose Antonio Sanz Universidad Publica de Navarra, Spain
REVIEW RETURNED	16-Apr-2019

GENERAL COMMENTS	Authors have accomplished most of my previous comments so the paper can be accepted now from my side.
---